# A Convolutional Neural Network for Impact Detection and Characterization of Complex Composite Structures

**DOI:** 10.3390/s19224933

**Published:** 2019-11-12

**Authors:** Iuliana Tabian, Hailing Fu, Zahra Sharif Khodaei

**Affiliations:** 1Department of Aeronautics, Imperial College London, London SW7 2AZ, UK; iuliana.tabian15@imperial.ac.uk; 2Wolfson School of Mechanical, Electrical and Manufacturing Engineering, Loughborough University, Loughborough LE11 3TU, UK; h.fu@lboro.ac.uk

**Keywords:** structural health monitoring (SHM), convolutional neural network (CNN), deep-learning, passive sensing, impact detection, impact characterization, composite structures

## Abstract

This paper reports on a novel metamodel for impact detection, localization and characterization of complex composite structures based on Convolutional Neural Networks (CNN) and passive sensing. Methods to generate appropriate input datasets and network architectures for impact localization and characterization were proposed, investigated and optimized. The ultrasonic waves generated by external impact events and recorded by piezoelectric sensors are transferred to 2D images which are used for impact detection and characterization. The accuracy of the detection was tested on a composite fuselage panel which was shown to be over 94%. In addition, the scalability of this metamodelling technique has been investigated by training the CNN metamodels with the data from part of the stiffened panel and testing the performance on other sections with similar geometry. Impacts were detected with an accuracy of over 95%. Impact energy levels were also successfully categorized while trained at coupon level and applied to sub-components with greater complexity. These results validated the applicability of the proposed CNN-based metamodel to real-life application such as composite aircraft parts.

## 1. Introduction

The next generation of aircraft have to comply with low carbon emission regulations. As structural weight is a key component in fuel consumption, it is natural to change from metallic to composite structures which are lighter. However, to fully utilize their strength to weight ratio, their conservative damage tolerance design needs to be revised. One way of ensuring the safety of the structure without overdesigning the components, is to have continuous monitoring of its state to detect any alarming external impact events
which could lead to loss of strength at early stages of evolution such as barely visible impact damage (BVID). Non-Destructive Inspection (NDI) techniques are commonly used to check the integrity of various engineering structures under operation to confirm their safe usage. There are various NDI methods that have been well developed for evaluating structural integrity as well as material characterization for different applications, such as bridges [1], composite structures [2] concrete foams [3].

However, not all exiting methods are suitable for detecting BVID in composite structures. In addition, the current NDI techniques face challenges to provide continuous monitoring of structures, specifically for parts without easy access. Structural Health Monitoring (SHM) techniques have gained a lot of attention in the last two decades with the aim of addressing theses shortcomings [4] by integrating large networks of sensors permanently onto the structure to monitor its integrity continuously in real-time.

There has been numerous research on the development and validation of SHM methodologies and technologies at laboratory environment, mostly on simple coupons. SHM of complex structures, such as composite lattice truss core sandwich structures [5] and composite honeycomb sandwich structures [6] have also been investigated and progressed.SHM techniques are usually divided into Impact detection and identification [7,8] and Damage detection and characterization [9,10]. The final goal of an SHM system is to carry out structural diagnosis and prognosis and provide the maintenance engineers with the required action and the remaining useful life of the structure. However, as the complexity of the structures increases, it will becomes more challenging for an SHM methodology to comply with the reliability requirements of an NDI technique. Usually for an aircraft this is 90%/95% i.e. 90% probability of detection (PoD) with 95% reliability.

In the era of internet of things (IoT), machine learning and data-driven techniques have become more attractive to be employed in cyber–physical systems such as SHM. These algorithms involve modelling complex relationships between the input and the output data, as presented in references [11,12,13]. One of the most frequently used methods for structural diagnosis based on sensor data, is Artificial Neural Network (ANN); examples can be found in references [14,15]. ANN is a machine learning algorithm that adapts its weights during the training phase. ANN has proven to work both on impact localization and force reconstruction, if the application is relatively simple and enough input data is given to it. Furthermore, the validity of the ANN metamodels directly depend on the training process and the amount of data available. Sensor data should not be input as a discrete signal as it contains too much information and it requires extraction of specific features such as Time of Arrival (ToA) of signals for impact detection on a plate, which is problem-dependant and cannot be generalized [16]. ANNs are generally accurate for the scope of a given training data, so for a real life impact identification and characterization, a large range of training data is required [17]. Another machine-learning method successfully employed for simple applications is the Support Vector Machine (SVM) which adapts its architecture automatically and requires less training data compared to ANN [16].

Other machine learning algorithms that have been tested for simple applications include: Extreme learning machines (ELM) [18,19], Probabilistic neural network (PNN) [20,21], Fuzzy ARTMAP network (FAN) [22,23,24], Least square support vector machines (LSSVM) [16] and others. However, these methods are very good for relatively-simple structures only, and they lack generalization. More recently, Convolutional Neural Network (CNN) has been the-state-of-the-art neural network in many fields [25], including image classification [26], object recognition [27], speech recognition [28,29], semantic segmentation [30], medical studies [31] and computer vision [32,33]. This is due to their outstanding performance, as well as readily available platforms for their implementation and open-source libraries capable of running Tensorflow, such as Keras. Moreover, the wide-spread deployment of low-cost sensors connected to the Internet, under the IoT evolution, facilitates data gathering at a much lower cost. The application of CNN in the SHM field has not gone unnoticed [34,35].

In reference [36] a new, real-time vibration-based structural damage detection system is proposed based on 1D Convolutional Neural Network. One of the main advantages of this method is that raw signals are used for the optimal damage-sensitive feature extraction. In another work [37], it is emphasized how CNNs can fuse and simultaneously optimise feature extraction and classification into a single task: a learning block in the training phase of the CNN, such that it eliminates the need of feature extraction beforehand. Thus, using a CNN with raw data as input will be more advantageous than traditional extraction methods. De Oliveira et al. [38] developed a CNN-based SHM technique for an Aluminium specimen for damage detection. Moreover, the rotating machinery domain features many research papers using CNNs such as [39], that uses images with the actual damage as training data [40,41,42,43,44], which emphasize that traditional methods ignore abundant information from the signals when extracting only a few features, such as mean value, standard deviation and kurtosis.

Although there have been recent advances in the technological developments for both active and passive sensing technologies [45,46], in terms of methodologies however, most of the reported work with CNN have been on simple structure with isotropic properties and the scalability of the method on real structures under operational load have not been demonstrated. The main aim of this work is to investigate the applicability of CNN in impact detection and identification of complex composite structures such as aircraft stiffened panel. In particular, the scalability of CNN-based metamodels is researched in order to propose a more realistic approach where the proposed methodologies will be developed on coupons or small scale structures and applied to real structural parts.

The contents of this paper is organized as follows: Section 2 provides the fundamental theory and essential architecture of CNNs. Section 3 discusses the system-level operation principle of the CNN and passive sensing-based methodology for impact detection and characterization, especially on how the passive sensing data can be prepared and used for CNN-based impact evaluation. Section 4 presents the application of the developed metamodel in a complex stiffened composite panel in terms of impact localization and impact energy characterization with the consideration of model up-scalability. Section 5 summarizes the main contribution and findings of this work with the discussion of potential future work.

## 2. Convolutional Neural Network

The Convolutional Neural Network is a deep-learning architecture that has been extensively studied in the last few years [47]. It is inspired by the natural visual perception mechanism of living creatures. The first CNN was the LeNet-5, by Le-Cun et al. [48], that was inspired by the discovery made by Kunihiko Fukushima that the neurons in the visual cortex are responsible for detecting light in receptive fields [49]. The CNN metamodel is a multi-class classification. The class or attribute represents the label associated with an image, based on which the network trains its weights. A class can be a specific location, a region, a group of locations, an energy level or any other.

In this paper CNN is used to construct a relationship between impact induced ultrasonic signals (recorded by permanently mounted sensors on the structure) and the output of the diagnosis which in this case is (1) impact detection, (2) impact location and (3) impact identification (energy level which can be related to its severity). A typical CNN architecture is shown in Figure 1, showing the configuration and the principle of using CNN for impact localization and classification. The proposed metamodel consists of a Convolution Neural Network algorithm that takes as input 2D images representing the raw data from the Piezoelectric (PZT) sensors, for different impact locations and energy levels. Then, it gives as output the class with the highest probability. First, the theory of CNN is briefly presented before reporting on its development and validation for the purpose of passive sensing.

### 2.1. Convolution Neural Network Theory

There are many variations in the CNN, but usually it has three types of layers: Convolution, pooling and fully-connected layers. The convolution layer is made of multiple convolution kernels that learn feature representations of the inputs and generate feature maps. A feature map is the result of first convolving the input with a learned kernel, and next, by applying on the convolved results an element-wise non-linear activation function. In mathematical terms, the feature value, $z_{i,j,k}^{l}$, at a certain location (i,j) in the $k^{th}$ feature map of the $l^{th}$ layer, is [47]:(1)$$z_{i,j,k}^{l}={w_{k}^{l}}^{T}x_{i,j}^{l}+b_{k}^{l}$$
with $w_{k}^{l}$ and $b_{k}^{l}$ being the weight vector and, respectively, the bias term of the $k^{th}$ filter of the $l^{th}$ layer. The kernel $w_{k}^{l}$ is shared between the $z_{:,:,k}^{l}$, feature maps, which is different from ANNs, for example. This had the advantage of reducing the model complexity and making it easier to train. If the non-linear activation function is called a(·), then the action value $a_{i,j,k}^{l}$ of convolutional feature $z_{:,:,k}^{l}$, is [47]:(2)$$a_{i,j,k}^{l}=a\left(z_{i,j,k}^{l}\right)$$

The advantage of the activation function is that it introduces non-linearities to CNN, that are good for detecting non-linear features. The most used activation function are sigmoid, ReLu and tanh. The pooling layer that usually is between two convolution layers, has the role of achieving shift-invariance by minimising the feature maps’ resolution. If the pooling function is pool(·), then for each feature map [47]:(3)$$y_{i,j,k}^{l}=pool\left(a_{m,n,k}^{l}\right),\forall \left(m,n\right)\in R_{ij}$$
with $R_{ij}$ being a local neighbourhood around location (i,j). Pooling operations can be average pooling and max pooling. Usually, the kernels in the first layers detect low-level features like curves or edges, and the kernels in higher layers are taught to encode more abstract features. The optional fully-connected layer at the end of the network has the role of taking all the neurons in the previous layer and connecting them to every single neuron of current layer, such that it generates global semantic information. The output layer is the last layer of a Convolutional Neural Network, and, usually, it is a Softmax operator or a SVM that is used.

The loss function, minimised during training, is [47]:(4)$$L=\frac{1}{N}\sum_{n=1}^{N}\left[l\left(\theta ;y^{\left(n\right)},o^{\left(n\right)}\right)\right]$$
where N is the number of the desired input-output relations $\left(x^{\left(n\right)},y^{\left(n\right)}\right),n\in \left[1,...,N\right]$ and $x^{\left(n\right)}$ is the $n^{th}$ input data, $y^{\left(n\right)}$ is its target label, and $o^{\left(n\right)}$ is the output of the CNN [47].

Convolutional neural networks have neurons that learn weights and biases, as other neural networks. CNN architecture assumes that the inputs are images, so certain properties are encoded into it. This improves the efficiency of the forward function and reduces the quantity of the parameters in the network. The difference from other neural networks is that the CNN have neurons arranged in three dimensions: width, height and depth, as displayed in Figure 1. The neurons in a layer are only connected to a small portion of the neurons from the previous layer, unlike in ANN.

### 2.2. Network Architecture


**The convolution layer** is the core of any CNN. It is used for extracting information from its input through the use of a number of filters that are automatically taught to detect certain features in an image. The filters’ size and their numbers is determined by the user. Each filter will scan through the input from the upper left hand side corner to the bottom right hand side corner, each creating a feature map. The neurons at the output are arranged in a volume with a depth equal to the number of filters, a height equal to $h_{i}-h_{f}+1$ (where $h_{i}$ is the input height and $h_{f}$ is the filter height) and a length equal to $l_{i}-l_{f}+1$ (where $l_{i}$ is the input length and $l_{f}$ is the filter length). As more convolutional layers are connected in series the output of one such layer becomes the input of another, and its features are extracted again increasing the level of complexity, and hence the accuracy, but also increasing the training time and the risk of overfitting [50,51]. Thus, there is a trade-off, and the number of convolution layers, as well as the number of filters and their sizes in the metamodel, were chosen by performing the trial-and-error method.**The pooling layer** performs the down-sampling in the width and height, reducing the dimensions of its input and, hence, reducing the number of parameters to be computed. This reduces the complexity of the network and the possibility of overfitting. The pooling operation operates on each depth slice of the input separately, down-sampling them all in the same manner. Each of the slices will be divided into a number of patches, equal in area to the filter size set by the user when defining the pooling layer. The most commonly used filter size is (2,2), so each slice will be divided into a number of adjacent but disjoint patches of 2 neurons high and 2 neurons long. The output of the pooling layer will be a smaller volume, but equal in depth to the input. For example, if the input to a pooling layer is 64 × 64 × 6 in volume and the filter size of the layer is (2,2), then the output is 32 × 32 × 6, achieving a great reduction in the complexity of the network. There are multiple types of pooling layers, categorized by the way this operation is performed. The most popular types are [50,51]:
—**Average Pooling**: Calculates the average of the numbers within each patch and sends it to the corresponding position in the output.—**Max Pooling**: For every patch, the maximum is sent to the output, as in Figure 2b. This type was shown to have a better performance [51], and was used in all the pooling layers of the CNN in this work.**The flatten layer** is used to change the shape of the input, making it an array of 1 neuron depth and height, equal in length to the product between the length, depth and height of the input to that layer. this layer is used in every CNN because the output layer must be a one-dimensional vector [50].**The dropout layer** is used to reduce overfitting by randomly cutting off a fraction of the nodes in the network. This random dropping of neurons in the network can be used to simulate a great number of different architectures which leads to a better generalization of the CNN [50].**The densely connected layer** is a regular fully connected layer. Each of its output neurons is connected to all the neurons from the input. This is usually implemented at the output together with a Softmax function to give the predictions. The nodes at the output of the layer, will, thus, contain the probabilities of the input to the CNN belonging to all classes. As each of those nodes is connected to all the neurons of the input to the layer, each receives all the information from the first half of the network, containing the convolutional and pooling layers. This means that the final prediction is made according to the whole input image, not just the output of some convolution or pooling filters [47,52,53].


### 2.3. Activation Functions

The activation function’s main purpose is to introduce nonlinearity in the relationship between the output of a node and the input of another node [54]. There are multiple types of activation function, and the main ones are described as follows:**Sigmoid function**: The curve has an ’S’ shape and it is given by the following equation:
(5)$$sig\left(x\right)=\frac{1}{1+e^{-x}}$$
Due to the fact that the function is not centred on the origin but on the (0, 0.5) point, as well as the limited region of high sensitivity, when using a sigmoid activation function, the learning algorithms will have difficulties in updating the weights in order to improve the performance causing a difficult process of optimisation and a slow convergence [50,54]. In addition, as the output varies between 0 and 1, if a large input is applied, it will be scaled down significantly. Therefore, a large change in the input will result in a small change in the output. This problem is called the vanishing gradient, and it can be problematic when using multiple layers in the network, the gradient can become very small and cause the weight and biases to not be updated very well [55].**Tanh function**: The hyperbolic tangent function is a slightly improved version of the sigmoid, in that the activation function is now centred on the origin. The function has an ’S’ shape, and will saturate at −1 for x=−∞, and 1 for x=∞. The function is given by:
(6)$$tanh\left(x\right)=\frac{2}{1+e^{-2x}}-1$$Using the tanh function, the optimisation will be easier, than for the sigmoid case. However, the output still saturates, the high sensitivity region is still small, and the vanishing gradient is still a problem [50,54]. The first derivative can be derived to be:
(7)$$\frac{d}{dx}tanh\left(x\right)=1-tanh^{2}\left(x\right)$$From Equation Equation 7, the relationship between the function and the first derivative is still simple, so it is easy for the function to be performed computationally.**ReLu function** (Rectified Linear Unit): Here, the function curve will have two regions, depending on the value of the input. For negative inputs, the function output is 0, while for positive inputs, the result is equal to the input itself:
(8)$$ReLu\left(X\right)=\left\{\begin{array}{cc} 0 & x<0\\ x & x\geq 0 \end{array}\right.=max\left(0,x\right)$$The ReLu function has numerous advantages when comparing with the sigmoid or the tanh activation functions. Firstly, it was proven to be approximately 6 times faster in convergence comparing to the hyperbolic tangent. Secondly, as the function increases from 0 to *∞* for positive inputs, a large variation in the input will be translated to a large variation in the outputs so the vanishing gradient problem is avoided. The function is no longer saturated and will have one non-linear region (i.e. for x<0) and one linear region (i.e. for x≥0), but overall it is still a non-linear function. Nevertheless, when using backpropagation for training the network, the linear region will bring many desirable advantages of linear activation functions. It is computationally easier performed than the previous two. On disadvantage of ReLu activation function, is that, for negative inputs, the function is horizontal, and, thus, the gradients will be zero. This means that, in that region, the weights will no longer be adjusted, causing a problem called dying ReLu resulting in a fraction of the network to become passive [56].**Leaky ReLu function**: This activation function is a version of the ReLu that does not have the dying ReLu problem [47]:
(9)LReLu(x)=max(x,0)+λmin(x,0).
where λ is a value between 0 and 1, set by the user. For negative inputs, the function will no longer exhibit a horizontal line, but it will allow a small non-zero gradient to exist, which will make the updating of the weights possible. Thus, a fraction of the network will no longer be passive. In the positive half, the LReLu will be identical to the ReLu. Therefore, all the aforementioned advantages of the rectified linear unit function can still be utilized [47]. Therefore, the Leaky ReLu is an improved version of the ReLu, and its application is investigated in this work.**Softmax Function**: This function is usually used for the output layer. It is used to normalise the output vector of the CNN, which is of length equal to the number of classes, say K, to a vector of length K, whose values sum to 1. This final vector will contain a range of probabilities, and the position of the maximum one will be the predicted class. The Softmax function was used during this project, too, and mathematically it can be written as [57]:
(10)$$f{\left(z\right)}_{j}=\frac{e^{z_{j}}}{\sum_{k=1}^{K}e^{z_{k}}}$$

### 2.4. Fitness Function

There are no general instructions on the optimum architecture of a neural network based metamodel. The most common way of finding the best structure is through trial and error. However, defining an appropriate fitness function which truly represents the accuracy and efficiency of the prediction is a fundamental step.


**Classification accuracy**
The classification accuracy is one way to evaluate the efficiency of the developed metamodel in predicting the output. It is defined as the percentage of the correctly predicted values from the total number of predictions [58]. This classification accuracy is useful, however, only when there are equal numbers of inputs belonging to each class [58]. Thus, another metric is needed, to be able to see how the code performs in predicting for each separate class.
**Confusion matrix**
Confusion matrix is a parameter which can quantify the performance of the metamodel for each class. The confusion matrix has a square shape, the number of rows & columns being equal to the total number of classes in the classification task. The sum of all the elements of column number *j* and *i* represents the total number of predictions for classes *j-1* and *i-1* respectively. In addition, the off-diagonal terms of the matrix show the wrongly predicted classes and the accuracy of the metamodel can be quantified easily, as shown in Figure 3.
**Loss function**
Another method of evaluating the performance of the algorithm is through the loss function. In machine learning, loss is applied as a penalty for a wrong prediction. This is important for the SHM application, since due to high safety factors, false alarm and mis-detection has to be minimum. For an exact prediction, the loss is zero, while inacurate categorization will result in greater loss. Therefore, the program will update the weights and biases until the loss is minimised. In a multi-label classification algorithm, logarithmic loss, also named cross-entropy loss, is commonly used according to [58,59]:
(11)$$\mathrm{Logarithmic}\;\mathrm{Loss}=\frac{-1}{N}\sum_{i=1}^{N}\sum_{j=1}^{M}y_{ij}\cdot \log p_{ij}$$
where $y_{ij}$ is either 1 indicating whether the sample number *i* belongs to class number *j* or 0 otherwise. In addition, $p_{ij}$, represents the probability of sample number *i* to be labelled by class number *j*, *N* is the total number of samples, and *M* is the total number of classes in the classification algorithm.

## 3. Passive Sensing Metamodel Based on CNN

The main objective of this work is to use CNN to develop a metamodel for passive sensing of composite structures, which results in impact detection and identification, i.e. location of the impact and magnitude of the impact energy to classify whether it is alarming or not, as presented in Figure 4. The impact events can be monitored and recorded by the distributed wireless sensor network in real-time. The recorded events will then processed and evaluated in the network coordinator with high computational performance using the CNN method with marginal delay.

An impact event will generate guided waves in the composite structure which will be recorded by permanently mounted piezoelectric (PZT) sensors, see Figure 5. This represents the Voltage of the signal as a function of time for the network of PZT sensors. Sensor location optimization is one of the key issues for SHM to obtain sufficient information for impact detection and localization. As the main focus of this paper is on CNN and passive sensing for impact detection, sensor location optimization methods will not be discussed, but are provided in Reference [60,61].

The input to a CNN is a 2D image, therefore, the discrete signals have to be processed into a right format for the training and development of the network. This is discussed in detail in the next section.

The convolution layers resemble the cells in the human visual cortex, therefore 2D images as inputs are the most appropriate. For this reason, an innovative method for transforming the PZT-recorded signals to 2D images have been proposed and implemented in this paper (see Figure 6). Subsequently, two CNNs are implemented, that take as input a number of *training* images and their associated attributes list, corresponding to multiple impacts. One CNN is for impact localization, and a separate one is for impact categorization (energy level). For impact localization, the structure is divided into sub-regions each representing a localization class while for energy level prediction, 3 classes are identified as safe, alert and damage based on the threshold of damage initiation defined for that part of the structure [45].

For the image generation, the raw data gathered from the PZTs for one impact is trimmed to eliminate the steady-state period to focus on the transient state features. Various lengths of input data were tested during for this work, with little or no impact on the performance, as long as the steady-state was eliminated.

### 3.1. Input Data Generation - Impact localization

The discrete signals recorded by the network of sensors is plotted as a surface map of the Voltage against time, for each sensor number, as illustrated in the example in Figure 7. The surface map in Figure 7a represents sensor signals for one impact locations of a set energy level on a curved stiffened panel. For each impact location different surface maps will be generated. These surface maps are then transformed to a 2D image by taking the top view and presented in grayscale to reduce the complexity of the image due to the RGB scale, see Figure 7b. As the signal is transferred to a colour scale image, the important information such as Time of Arrival (ToA) and amplitude of the signals are maintained. This is very important as the ToA of the signals are directly linked to the localization of impact and amplitude of the signals reflect the level of impact energy.

However, impacts of different energy will result in different signal amplitude. The aim of this work is to develop a metamodel which is scalable to different structures or parts of the structure with similar geometrical features. Therefore, for the purpose of impact localization, the emphasis of the training has been on the relationship between the signal amplitude and proximity of the sensor to the impact location, and not the influence of impact energy on the sensor signal. This will ensure generalization of the trained neural network to predicts impacts of magnitude which have not been used in the training phase and the possibility of scalability of the developed network to larger structures. Therefore, for each impact, the signals recorded by the sensor network are normalized by the highest amplitude of the signal to maintain the same gray scale image for each input. The input image will then change with different impact location only. For example see the input images in Figure 8a,c where the impact location is the same but the impact height in Figure 8c is doubled (and hence the energy) but input images are identical. Same can be seen for a second impact location in Figure 8b,d where the variation of colour distribution in each image correspond to how far the impact is from each sensor. This is one of the novelties of the proposed metamodel which means that the location prediction is independent of the energy level of the impact as distinct levels of energy, this method ensures that the energy level of the impact does not have a contribution to the location prediction of the impact. Consequently, the same input image cannot be used for prediction of the impact energy levels. These figures show a relative intensity. Therefore, no absolute value was associated with each colour intensity, which means that a colour bar is not necessary for additional information in impact localization.

### 3.2. Input Data Generation-Impact Energy Level

To have the best response from a non-linear metamodel, the input and output should be physically linked. Therefore, to find the optimum input to predict energy of an impact, several features have been investigated in this work and two novel methods are introduced to generate the input images from sensor signals which are related to energy levels. The first one favours transferability to other applications, while the second one incorporates a physical meaning into the images.

#### 3.2.1. Transferred Energy

The first method consists of integrating the area under the absolute value of the Voltage, for each sensor reading to measure the transferred energy in the plate due to each impact. The original signal is shown in Figure 9a, while the bottom image shows the area under the absolute value of the signal in time domain. The area corresponding to each sensor readings is plotted as a bar plot in Figure 10a,b for two impacts of different energy but same location.

The above images illustrate the differences in the magnitudes for distinct energy levels. These are, subsequently, used as input to the CNN-based metamodel for the impact energy prediction. This method is intended to be simple, such that the metamodel can be easily transferred to many other applications and experiments, without the need to have expert knowledge in Signal Processing.

#### 3.2.2. Instantaneous Energy and Averaged Stored Energy Method

Another method which has stronger physical link between the input and output in terms of categorizing the levels of impact energy as low, alert and high is presented here which was tested to give similar results for all the impact levels. The piezoelectric sensors can be represented as capacitors in parallel to a current source, so the Instantaneous Energy that the PZT sensor generates can be calculated as in [62]:(12)$$E_{in}=\frac{1}{2}C_{\mathrm{PZT}}V^{2}$$
with V being the output voltage, and $C_{\mathrm{PZT}}$ the capacitance of the PZT sensor. The Instantaneous Stored Energy for two impacts of different energies is shown in Figure 11a,b.

The Averaged Stored Energy, $E_{avg}^{j}$, can be defined as:(13)$$E_{avg}^{j}=\frac{1}{N}\sum_{n=1}^{N}E_{in}^{ij}$$
with j = 1,2…8 being the channel number and N: the total number of samples recorded per sensor.

Figure 12a,b show the Averaged Stored Energy for each of the sensors, for two impacts of distinct energy levels.

### 3.3. CNN Architecture

The compilation of the CNN algorithm is made up of two or three **phases**:**Training**, in which the initially random weights are adapted by passing the *training images* in batches, back and forth inside the network, to minimise a pre-defined loss function.**Validation (optional)**, which is used for optimising the network architecture. However, as the number of images per class was quite small for many of the applications discussed in this work, the dataset could not be split into three groups, so no validation was used.**Testing**, in which the generalization of the network is assessed and an output of predicted classes is given to the set of *testing* images.

After the training phase is done, the weights of the network can be saved. This allows to use the CNN for classification of new images, without the need to retrain, making it very quick.

All the results that will be presented during this project represent the average value of the multiple runs’ results. Usually, the CNN is tested 20 times for the same test case, but if there is not much variation in the results, 10 times are enough, too. When extracting the performance metrics for each test case, it was ensured that convergence was reached, for both accuracy and loss, and the loss was decreasing towards 0. Moreover, the loss must be at least under 1, if not under 0.5.

## 4. Application of the Metamodel to a Composite Stiffened Panel

In this section, the proposed metamodelling technique based on CNN is applied to a composite stiffened panel shown in Figure 13, to localize and characterize impact events.

### 4.1. Experimental Set-up and Data Acquisition

The composite part consists of a fuselage section, with stiffeners and frames and 12 DuraAct PZT sensors attached on the inside. The panel is a 1150 × 750 mm with a radius of curvature to the outer surface of 1978 mm. The skin is 1.66 mm thick with [45/−45/90/0/90/0/90/−45/45] layup and the stiffeners are 1.29 mm thick [45/−45/0/90/0/−45/45] layup. The skin and omega hat stiffeners were made of T800/M21 uni-directional pre-pregs [62].

The PZT sensors are connected to the NI-PXIe-1073, to record the sensor response. The impactor is a steel hemispherical impactor as described in [17]. The dropping height was varied in steps of 20 mm from 20 to 80 mm. The impacts were induced on 9 locations along the panel, shown in Figure 14b as squares. Both the sensors and the impact locations are inside the region delimited by the two frames. The locations and the sensors have been named such to ensure symmetry between top and bottom half parts of the panel. The data was recorded at a sample rate of 250,000 per second, for a total of 10,000 samples to avoid unnecessarily high data samples. Data was collected from the sensors during the impact experiment.

As the panel is curved, even though the impact set up is designed to impact perpendicular to the panel, there is a certain degree of tolerance in the impact location and angle which is good for testing the regularization capability of the developed metamodel.

Dataset D consists of impacts on 9 locations as seen in Figure 14b, from 4 different heights, ranging from 20 to 80 mm, repeated 4 times for each case. For each impact, data was recorded simultaneously for all sensors.

When generating the images for the CNN, the same techniques described in Section 3.1 and Section 3.2 were employed. Initially, signals from all the sensors have been used as input, but, subsequently, it was observed that the same performance if not better is achieved by using data from the 4 sensors which are closest to an impact event. Therefore, a two-step data processing methodology was proposed, where in the first step, based on the amplitudes of the received signals (if above a set threshold which corresponds to impact detection), the four sensors which are closest to the impact event are chosen. Only the transient part of the signals were used and arranged in a grayscale image to reflect the proximity of the impact generated signals to each sensors, see examples presented in Figure 15.

### 4.2. Impact Location Prediction

#### 4.2.1. CNN Architecture

The CNN Architecture was optimised for running with very small datasets, and the network architecture that performed the best for this application was chosen, see Figure 16.

Several modifications were made to be able to handle small datasets without overfitting: The first change is to eliminate one of the two Convolutional layers from each pair, as to decrease the number of parameters that needs to be calculated. This is because, for a small dataset, the strategy for the CNN is to learn few enough features not to overfit. The second modification was to add Dropout after each Convolutional layer, as Dropout is one way to prevent overfitting [63].

#### 4.2.2. Results

The Table below summarises the test cases for the location prediction of impacts on the stiffened panel. Dataset D is summarised in Table 1. The focus of the metamodel has been not to localize impacts with high spatial accuracy, but to with high reliability and decision making accuracy with minimum required data which is more desirable in application to real structures.

The D1 test case corresponds to a total of 96 total images, divided into 3 classes, specifically: left, middle and right hand sides of the panel (Figure 17).

The CNN was trained with 3 sets of data corresponding to each of the 6 locations (top and bottom parts of the panel), with data recorded from 4 sensors, for 4 different energies. Subsequently, it was tested with another set, following the same structure as the training set, but with newly acquired data. The accuracy is 100 %, meaning that the network identified every impact case (after detection) and localized it at the left, middle or right side of the panel. The network performance can be represented by the plot of the confusion matrix in Figure 18.

#### 4.2.3. Symmetry and Up-Scalability

The D2 test case consists of training data of impacts on the top part of the panel only, and testing the CNN performance with data of impacts on the bottom part. This checks if the network can be up-scalable and if it can generalize by training it for part of the structure and applying it to other similar sections. By training with 49 images and testing with another 49 sets of data, an accuracy of 99.4 % was achieved in classifying the correct impact localization class. This is a very important feature as it means scalability of the method to larger structures without having to train each section, subject to symmetry of the part.

### 4.3. Energy Prediction

#### 4.3.1. CNN Architecture

The network used for energy prediction, after testing various architectures is chosen as the one with the best performance and no overfitting, shown in Figure 19. The two main features of this CNN are: the Leaky ReLu activation functions was added after each Convolution Layer and only 2 sets of Convolution-Pooling layers were retained. One reason to use the Leaky Relu activation function (rather than Relu) was that the network was seen to face the problem called dying ReLu, in which several neurons do not respond anymore resulting in a fraction of the network becoming passive. Therefore, to solve the dying ReLu problem, this was replaced by the Leaky ReLu activation function, which stopped the architecture getting blocked and increased the accuracy. This can be seen in Table 2 by comparing the D4 (Relu) and D5 (LEaky Relu) sets.

#### 4.3.2. Results

The accuracy of the impact energy classification CNN is at 100%, as seen in in Table 2 for the D5 test results. The D6 test was performed using 72 images for training, representing 3 sets of impacts, and another set of 24 images for testing. The accuracy obtained is 98.3 %, at which the network accuracy converged very early, at about 10 epochs, as seen in Figure 20a. Figure 20b shows the loss converging to 0 very quickly, too. The D6 was run with images from 6 locations at the same time, which are the L1 to L6 in Figure 14. It is worth mentioning that throughout this work, for every metamodel that has been developed, the performance of the metamodel has been assessed on a new set of data for testing and the test dataset was never introduced to to the network during the training and validation phase.

Figure 21a represents the confusion matrix, in which it can be observed on the main diagonal that all impacts, except one, were correctly predicted. Therefore, the metamodel correctly divided all the impacts into 4 regions: *Safe, Warning, Alert, Danger*, as exemplified in Figure 21b, except one impact, which was wrongly categorized as Alert instead of Warning. In real applications, the threshold values for these four categories will be measured experimentally, for the specific material and composite lay-up and the range will cover larger energy levels.

## 5. Conclusions

A novel metamodel based on Convolutional Neural Networks and passive sensing for impact detection and characterization in composite plates was successfully developed, tested and optimised. The applications of the metamodel on a complex composite stiffened panel (stiffeners, curvature, frames) was conducted with the consideration of network architecture optimization, complexity and up-scalability, showing excellent levels of accuracy.

The metamodel accuracy reached values between 94.3% and 100 % when predicting distinct impacts on similar locations that it has been trained with. Moreover, the metamodel up-scalability was demonstrated by testing with impacts on locations out of the training region. For this case, the prediction accuracy was over 95%, with most of the cases being over 99.4%.

Regarding the energy level, the strategy was to classify the impact energy into distinct energy categories, such as *Safe*, *Alert*, *Danger*. The predictions reached an accuracy of over 98.3%, showing that it is a reliable method of classifying the impacts depending on the level of risk. The thresholds of the risk categories can be established from experiments, or can be based on material properties.

An innovative methodology was proposed for transforming the raw data acquired from a network of Piezoelectric sensors, to 2D images which can be input into a CNN. Another novelty of the work was to show that the metamodels could be developed and trained with minimum data. Despite that it is widely believed that CNNs require huge amount of data, the datasets used in the experiment were not larger than 280 impacts, which were easily acquired in the laboratory. For instance, for the stiffened panel, only 96 sets of data were used, giving a very good accuracy. This showed that a CNN-based metamodel for passive sensing on composite structures works remarkably, even for low datasets. Methods for addressing the main issue of CNNs: overfitting, were presented and implemented.

The advantages and the disadvantages of the metamodel were explored during this work. The strengths of the metamodel are: scalability to real-life applications; the ability to learn the extraction of optimal features automatically from unprocessed data; and the transferability to other applications and complex structures. However, this method still faces issues and challenges in setting appropriate rules in optimizing this metamodel automatically, selecting sufficient length of input data and generalizing this model using larger sets of training data.

In conclusion, this work is a proof of concept, and future work will be dedicated to impacts of higher energies and the effect of operational and environmental effects on the accuracy of the prediction. Reliability and probability of detection need to be addressed as well to comply with non-destructive inspection requirements. This proposed metamodel will be one of the enablers to achieve the goal of autonomous structural integrity inspection combined with wireless sensor networks in the era of condition-based maintenance.

## Figures and Tables

**Figure 1 sensors-19-04933-f001:**
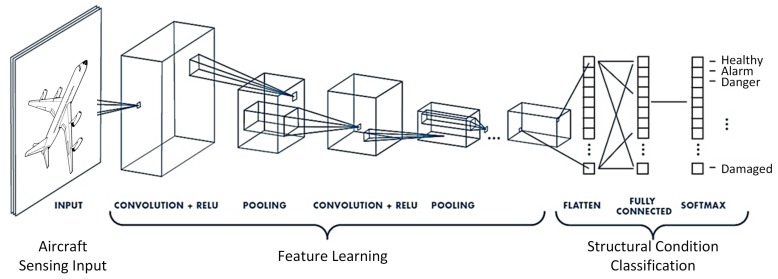
A typical Convolutional Neural Network architecture in aircraft structural health monitoring; the typical configuration, inputs and outputs are illustrated showing the system operation principle.

**Figure 2 sensors-19-04933-f002:**
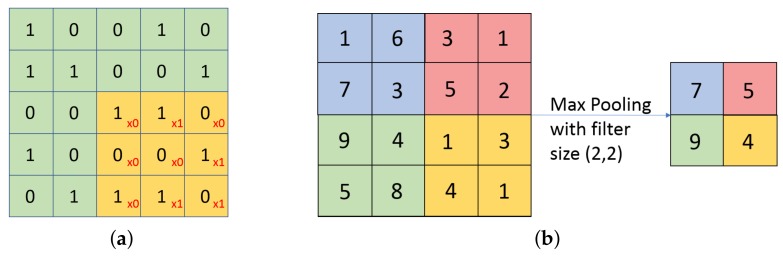
(**a**) Convolution operation. The amber squares represent the position of the kernel as it slides through the green input slice. (**b**) Max Pooling Operation with a filter size of (2,2).

**Figure 3 sensors-19-04933-f003:**
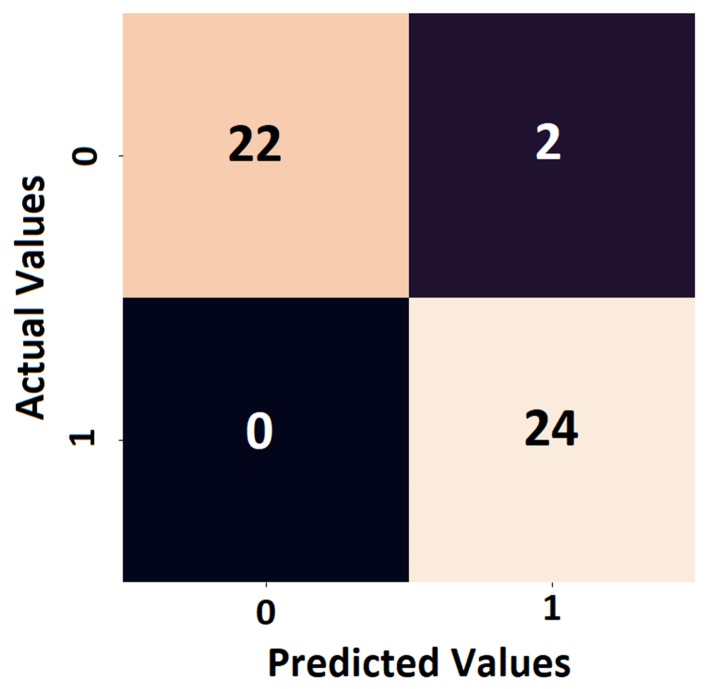
Example of a confusion matrix which shows that class *0* had 22 correct predictions and 2 wrong predictions of class *1*, while class *1* had all 24 samples predicted correctly.

**Figure 4 sensors-19-04933-f004:**
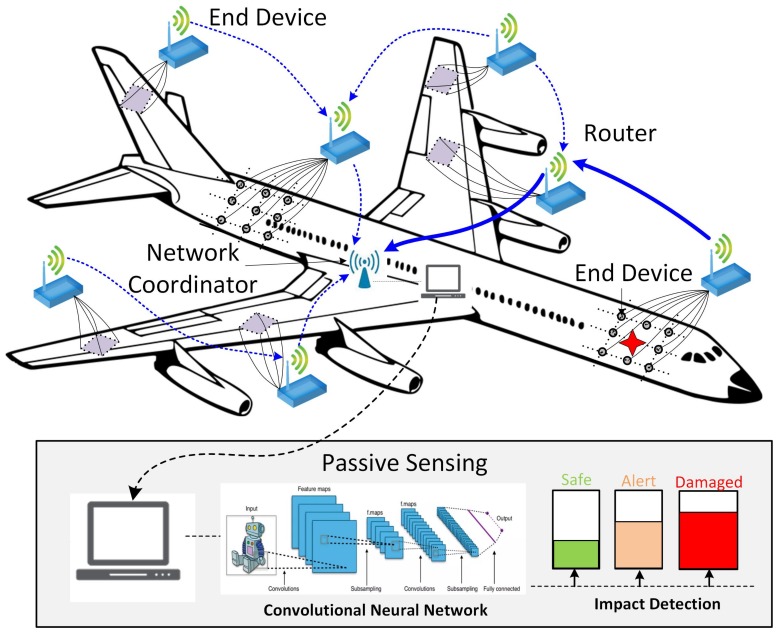
Passive sensing with embedded wireless sensor networks and CNN. Wireless passive sensing devices are mounted on an aircraft. A wireless sensor network is established to fulfil the impact detection, data communication and signal processing functions.

**Figure 5 sensors-19-04933-f005:**
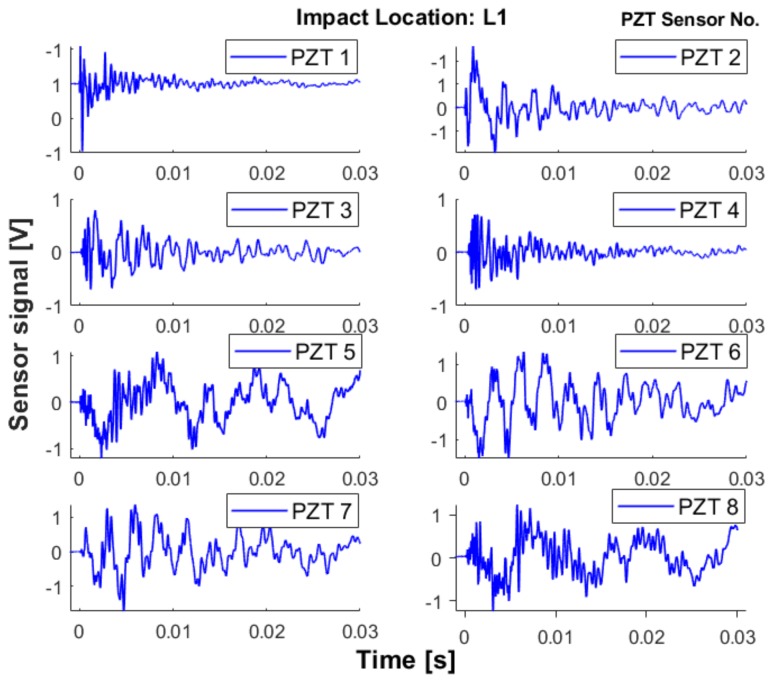
Example of multi-channel recording of impact signals at a certain location on a composite plate.

**Figure 6 sensors-19-04933-f006:**
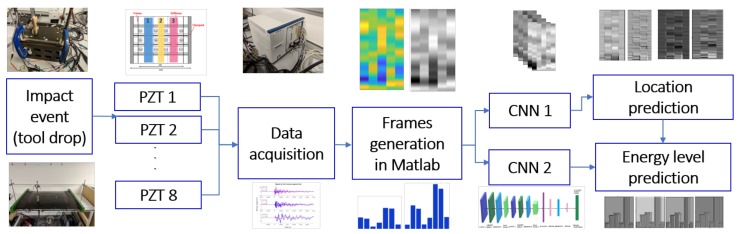
The methodology, from impacting the coupon, to data acquisition and impact identification and characterization.

**Figure 7 sensors-19-04933-f007:**
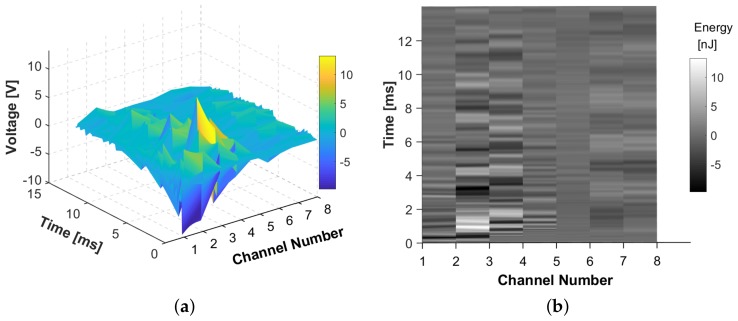
Example of (**a**) surface map of the Voltage recorded by the PZT sensors and (**b**) the 2D view in grayscale.

**Figure 8 sensors-19-04933-f008:**
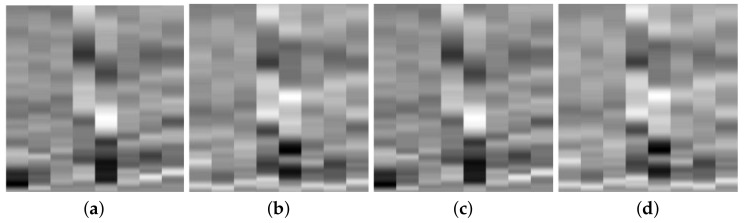
Example of input to CNN for impact localization: (**a**) Impact at location 1 and energy level 1, (**b**) location 2 energy level 1, (**c**) location 1 energy level 2 and (**d**) location 2 energy level 2.

**Figure 9 sensors-19-04933-f009:**
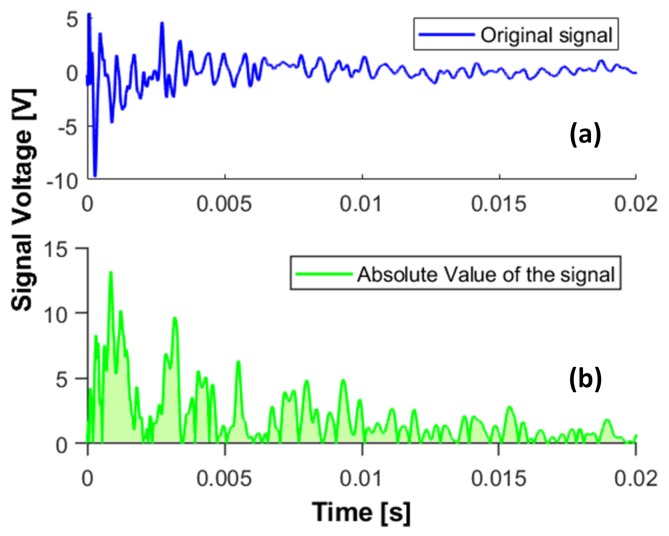
(**a**) The original signal recorded by one of the PZTs. (**b**) The hatched area underneath the signal is used for creating the bar plots in Figure 10.

**Figure 10 sensors-19-04933-f010:**
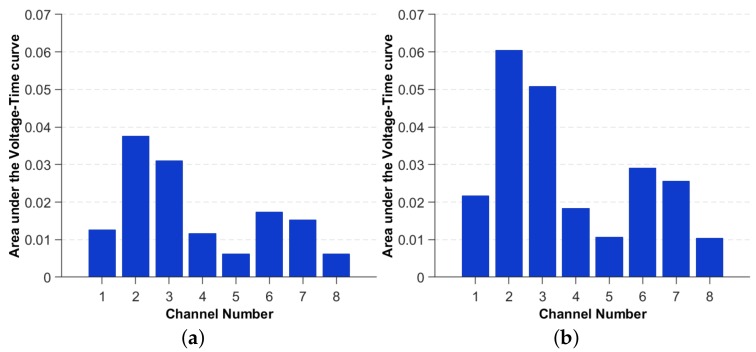
Example of bar plots of transferred energy for two impacts, same location and two energy levels (**a**) 49 mJ and (**b**) 98 mJ, respectively.

**Figure 11 sensors-19-04933-f011:**
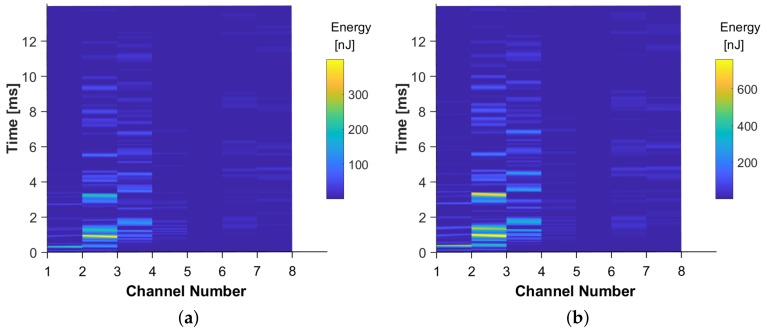
The Instantaneous Generated Energy against time, for 8 channels, for two different energy levels (**a**) 49 mJ and (**b**) 98 mJ, respectively.

**Figure 12 sensors-19-04933-f012:**
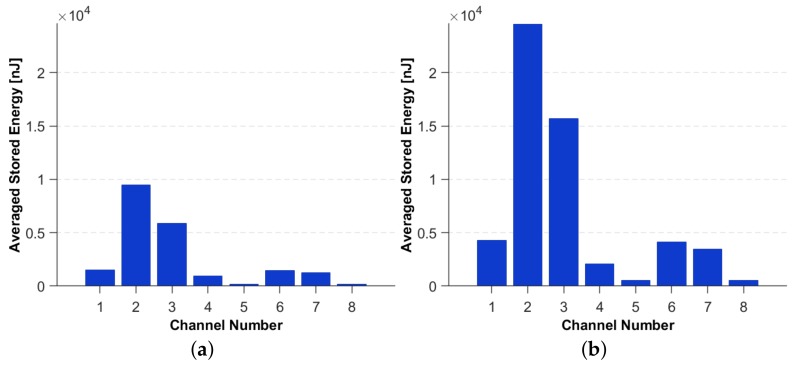
The Averaged Stored Energy for (**a**) 49 mJ and (**b**) 98 mJ impact energy.

**Figure 13 sensors-19-04933-f013:**
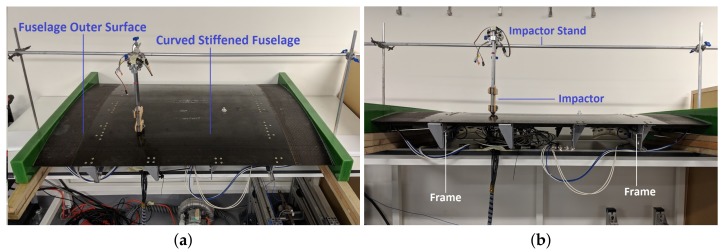
Experimental set-up: (**a**) A stiffened composite fuselage panel with the impactor fixed on an adjustable stand, (**b**) side-view of the panel.

**Figure 14 sensors-19-04933-f014:**
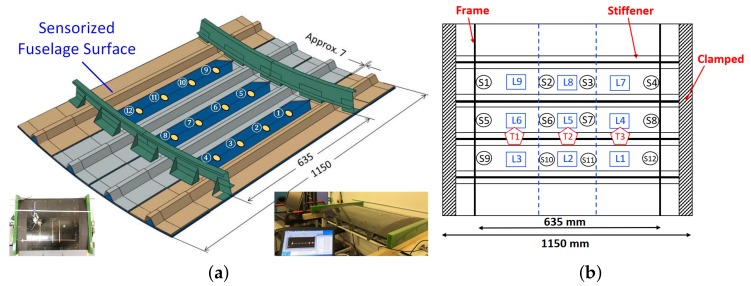
(**a**) Composite fuselage model with stiffeners, frames and surface-mounted PZT sensors. (**b**) The 12 PZT sensors configuration represented with a circle.

**Figure 15 sensors-19-04933-f015:**
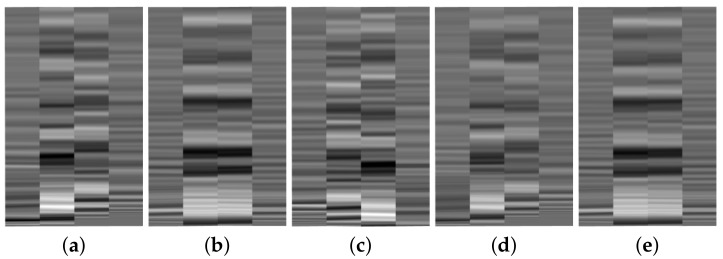
Images used for training the CNN for predicting the location of the impact. From left to right, the images correspond to locations 1, 2, 3, 7, 8 on the panel in Figure 14b.

**Figure 16 sensors-19-04933-f016:**
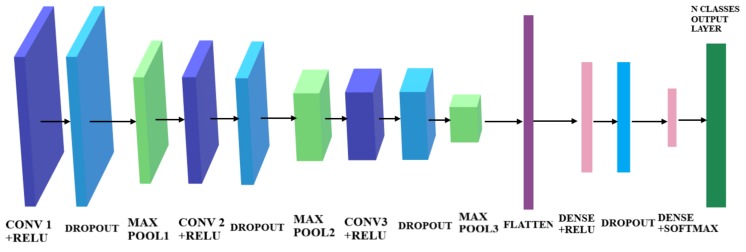
The proposed CNN architecture containing 3 repeating sets of Convolutional layer, Dropout, Pooling, followed by a Flatten, Dense, Dropout, Dense layers.

**Figure 17 sensors-19-04933-f017:**
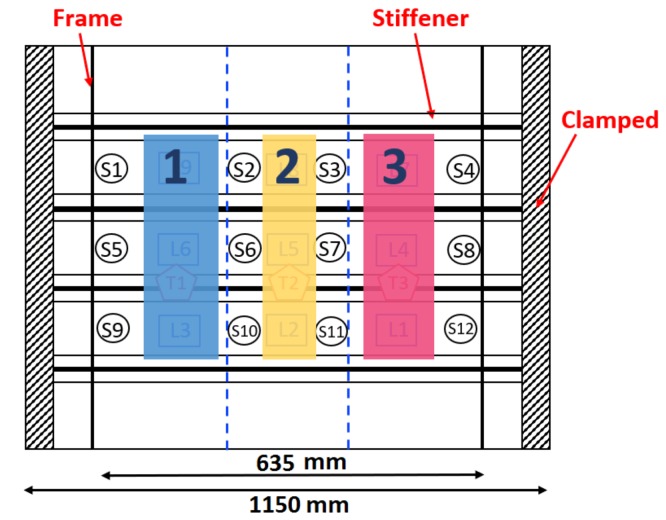
The assignment of classes for the impact locations on the stiffened composite fuselage model.

**Figure 18 sensors-19-04933-f018:**
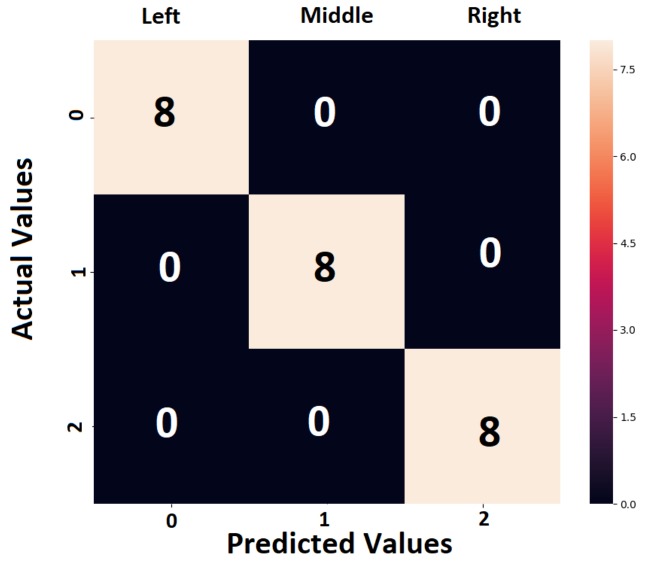
The confusion matrix corresponding to D1 case, in which the classes were: left (0), middle (1), right (2).

**Figure 19 sensors-19-04933-f019:**
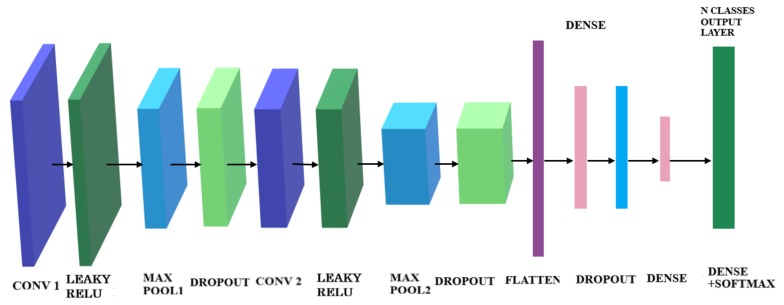
Metamodel architecture based on Leaky Relu activation function.

**Figure 20 sensors-19-04933-f020:**
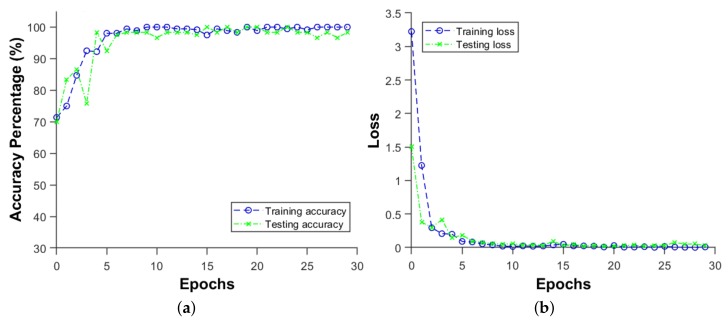
(**a**) Accuracy and (**b**) loss of the energy prediction for the stiffened panel for the D6 run in Table 2.

**Figure 21 sensors-19-04933-f021:**
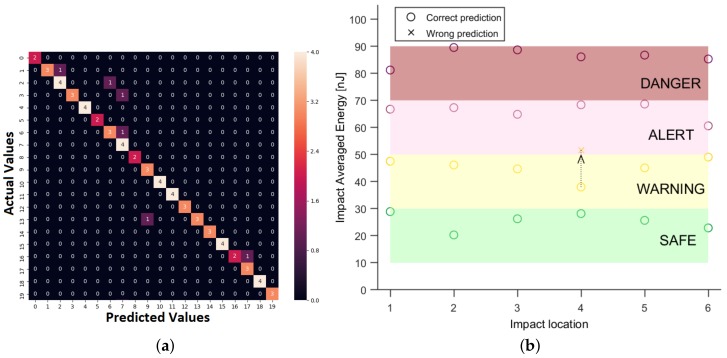
(**a**) The confusion matrix corresponding to run D6 in the Table 2. Each class represents a range of energy levels. (**b**) Energy level prediction for 6 locations, for 4 distinct energy levels.

**Table 1 sensors-19-04933-t001:** The results for the location prediction for impacts on the stiffened panel.

Name	Dataset	Total No. of Images	No. of Sensors	Training Data	Training Data Details	Testing Data	Testing Data Details	Classes	Images Per Class	Epochs	Accuracy (%)
D1	D	96	4	72	3 sets	24	1 set	3	24	30	100
D2	D	98	4	49	**Top** (L&R)	49	**Bottom** (L&R)	3	16–17	30	87.3
D3	D	98	4	49	**Top** (L&R)	49	**Bottom** (L&R)	3	16–17	30	99.4

**Table 2 sensors-19-04933-t002:** The energy prediction results for the stiffened panel.

Name	Dataset	Total No. of Images	No. of Sensors	Training Data	Training Data Details	Testing Data	Testing Data Details	Classes	Images Per Class	Epochs	Accuracy
D4	D	98	4	49	Top	49	Bottom	2	24–25	30	96.1
D5	D	98	4	49	Top	49	Bottom	2	24–25	30	100
D6	D	98	8	72	3 sets	24	1 set	4	18	30	98.3

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
