# Peer review of "A Convolutional Neural Network for Impact Detection and Characterization of Complex Composite Structures"

_sensors, 2019, doi:10.3390/s19224933_

Round 1

Reviewer 1 Report

The paper presents a meta-model for impact detection, localization and characterization of complex composite structures based on Convolutional Neural Networks (CNN). The ultrasound signal generated by impactor are recorded and converted to 2D images for characterisation and model validation.

The paper is very well written and the methodology is described extensively. However, the introduction should be improved to better describe the application of non-destructive testing (NDT) for structural health monitoring and material characterisation. The reviewer recommends including the following papers to support the NDT introduction.

Maizuar, Maizuar, et al. "Detecting structural damage to bridge girders using radar interferometry and computational modelling." Structural Control and Health Monitoring 24.10 (2017): e1985.

Liu, Lizhao, Saeed Miramini, and Ailar Hajimohammadi. "Characterising fundamental properties of foam concrete with a non-destructive technique." Nondestructive Testing and Evaluation 34.1 (2019): 54-69.

Maizuar, Maizuar, et al. "Structural Health Monitoring of Bridges Using Advanced Non-destructive Testing Technique." ACMSM25. Springer, Singapore, 2020. 963-972.

Please provide a short description of the Convolutional Neural Network architecture in Figure 1 caption.

Please provide a short description for the Passive sensing methodology in Figure 4 caption.

Please clarify whether this technique will be used for real-time impact detection. If yes, please describe how the technique can be potentially used in Aircrafts.

Author Response

The authors would like to thank the reviewers for their time, the provided comments and corrections which has made the paper improve. The authors have now addressed all the reviewers’ comments and applied to corrections in the revised version of the paper. We hope that with these improvements the paper will be in a state to be accepted for publication.

Kind regards

Iuliana Tabian, Hailing Fu and Zahra Sharif Khodaei

Response to Reviewer #1:

The paper is very well written, and the methodology is described extensively. However, the introduction should be improved to better describe the application of non-destructive testing (NDT) for structural health monitoring and material characterisation. The reviewer recommends including the following papers to support the NDT introduction.

Reply: The authors appreciate the reviewer’s comments on our work. The changes have been provided accordingly. A description for non-destructive testing (NDT) has been added in the early part of this paper with the references. As the paper focuses more on Structural Health Monitoring (SHM) and machine learning, the detailed progresses on NDT are not provided.

Changes: First paragraph on Page 1. Marked in RED.

Please provide a short description of the Convolutional Neural Network architecture in Figure 1 caption.

Reply: A description has been added in the caption as suggested.

Changes: “A typical Convolutional Neural Network architecture in aircraft structural health monitoring; the typical configuration, inputs and outputs are illustrated showing the system operation principle.”

Please provide a short description for the Passive sensing methodology in Figure 4 caption.

Reply: A description has been added in the caption as suggested.

Changes: “Passive sensing with embedded wireless sensor networks. Wireless passive sensing devices are mounted on an aircraft. A wireless sensor network is established to fulfil the impact detection, data communication and signal processing functions.”

Please clarify whether this technique will be used for real-time impact detection. If yes, please describe how the technique can be potentially used in Aircraft.

Reply: Yes, a real-time impact detection system is achievable by the proposed system in this paper. The wireless sensor network described in Figure 4 can monitor and record impact events and send recorded events to the network coordinator that have high computation performance suitable for running CNN algorithms. Delays due to wireless communication and running CNN testing functions will be unavoidable but are marginal.

Changes: First paragraph in Section 3, marked in RED.

Reviewer 2 Report

Please find enclosed my report.

Author Response

The authors would like to thank the reviewers for their time, the provided comments and corrections which has made the paper improve. The authors have now addressed all the reviewers’ comments and applied to corrections in the revised version of the paper. We hope that with these improvements the paper will be in a state to be accepted for publication.

Kind regards

Iuliana Tabian, Hailing Fu and Zahra Sharif Khodaei

Response to Reviewer #2:

The authors appreciate the reviewer’s positive comments and detailed suggestion on improving the paper. Changes have been made and listed below.

Please use the same spelling for the word “characterization”, “localization”, “categorized”, “generalization”, “emphasize” and so on throughout the article.

Reply: The authors have checked the whole manuscript on these suggested words. Now the same spelling rules are applied. Changes have been marked.

Line 34: please insert the acronym “Internet of Things (IoT)” and not in line 59.

Reply: This has been revised and marked.

Lines 36, 38 and 61: please insert ‘in References…’

Reply: This has been revised and marked.

Lines 47 to 49: please correct the grammatical structure of the sentence.

Reply: This grammatical issue has been revised and marked.

Line 68: please correct the typo: “Aluminium”.

Reply: This has been checked, and the authors believe the current spelling is appropriate.

Line 190: please correct the typo: “utilized”.

Reply: This has been corrected and marked.

Figures 7, 8 and 15: why are not the figures in colour?

Reply: The figures are plot in grey to simplify the complexity of the information given to the CNN. It will provide the algorithm with a limited variation in colour and increase the computation accuracy. This is also provided in Section 3.1, as marked in RED.

Figures 8 and 15: it would be useful to put the colour bar.

Rely: These figures show a relative intensity. The values are normalized using the maximum value in each set of data for one impact. Therefore, no absolute value was associated with each colour intensity, which means that a colour bar would not really provide additional information to readers or the CNN algorithm. Further explanation is provided in the paper, marked in RED above Figure 7.

Legend of Figure 13: please correct the typo: “stand” and not “atand”.

Reply: This has been corrected and marked.

Lines 219, 241 and 369: please insert “in Figure…”

Reply: This has been checked and marked.

Line 284: please correct the typo: “piezoelectric”

Reply: This has been checked and marked.

In subsubsection 4.3.2, one should not start with a results table but better introduce and explain it first.

Reply: The position of the table has been adjusted.

Please use the same spelling for the word “metamodel” throughout the article.

Reply: This has been checked and updated.

In the conclusion (line 417): “The advantages and the disadvantages of the metamodel were explored during this work”. It may be necessary to recall them succinctly in the conclusion.

Reply: This discussion has been added and marked in RED in the conclusion.

In the conclusion, the application perspectives and limitations of the method should be explained more.

Reply: This limitation is discussed in the disadvantages of this metamodel in the conclusion. This application perspectives and future work have been included in the last paragraph in the conclusion. Changes have been marked in RED.

Reviewer 3 Report

In this paper, a meta-model based on CNN is developed for impact detection, localization and characterization of complex composite structures. The scalability and applicability of the proposed technique are systematically investigated and verified. In general, this manuscript is well written and the approaches are solid. I recommend its publication in Sensors after addressing my following minor comments.

1. In introduction part, it is mentioned that most of the existing SHM and NDI techniques are reported for simple structure. Actually, kinds of methods have been proposed for complex composite structures, such as composite lattice sandwich structures (Composite Structures, 2015, 126: 34-51.), composite honeycomb sandwich structures (Mechanical Systems and Signal Processing, 2016, 76: 497-517.) etc., it is suggested to discuss a little bit of these work to make the background more comprehensive.

2. A series of PZT sensors are used in the proposed technique, and the effect of PZT number on the detection result is discussed. Can the authors give a suggestion on the arrangement sparsity of PZTs to obtain a good detection accuracy?

3. Page 7, modify “the off-diagonal terms of the matrix shows…” as “the off-diagonal terms of the matrix show…”. In addition, the font size of the color bar in Fig. 3 is too small.

4. Page 8, “as presented in Figure 4”.

5. Page 11, Fig. 9. It is suggested to use “(a)” and “(b)” to replace “Top” and “Bottom”.

Author Response

The authors would like to thank the reviewers for their time, the provided comments and corrections which has made the paper improve. The authors have now addressed all the reviewers’ comments and applied to corrections in the revised version of the paper. We hope that with these improvements the paper will be in a state to be accepted for publication.

Kind regards

Iuliana Tabian, Hailing Fu and Zahra Sharif Khodaei

In introduction part, it is mentioned that most of the existing SHM and NDI techniques are reported for simple structure. Actually, kinds of methods have been proposed for complex composite structures, such as composite lattice sandwich structures (Composite Structures, 2015, 126: 34-51.), composite honeycomb sandwich structures (Mechanical Systems and Signal Processing, 2016, 76: 497-517.) etc., it is suggested to discuss a little bit of these work to make the background more comprehensive.

Reply: The authors appreciate the reviewer’s suggestions on improving the paper. Discussion and references on SHM of complex structures have been added and marked.

Changes: The 2nd paragraph in Section 1. Marked in RED.

A series of PZT sensors are used in the proposed technique, and the effect of PZT number on the detection result is discussed. Can the authors give a suggestion on the arrangement sparsity of PZTs to obtain a good detection accuracy?

Reply: Senor location optimization is one of the key issues for SHM. There have been some research articles in the literature. The authors’ research group has also conducted some relevant research on this topic. As the focus of this paper is on CNN and passive sensing for impact detection. The detailed sensor location optimization will be not included, but relevant references have been provided.

Changes: Marked in the 2nd paragraph in Section 3.

Page 7, modify “the off-diagonal terms of the matrix shows…” as “the off-diagonal terms of the matrix show…”. In addition, the font size of the colour bar in Fig. 3 is too small.

Reply: The grammar is corrected. The colour bar in Figure 3 has been removed, as it will not provide any necessary information for the confusion matrix.

Page 8, “as presented in Figure 4”.

Reply: This has been revised and marked.

Page 11, Fig. 9. It is suggested to use “(a)” and “(b)” to replace “Top” and “Bottom”.

Reply: This has been revised in the figure and the caption.

Round 2

Reviewer 1 Report

All of the comments are addressed and the manuscript is modified.